# Characterizing the Anoxic Phenotype of *Pseudomonas putida* Using a Bioelectrochemical System

**DOI:** 10.3390/mps2020026

**Published:** 2019-03-30

**Authors:** Bin Lai, Anh Vu Nguyen, Jens O Krömer

**Affiliations:** 1Systems Biotechnology group, Department of Solar Materials, Helmholtz Centre for Environmental Research—UFZ, 04318 Leipzig, Germany; bin.lai@ufz.de (B.L.); anh-vu.nguyen@ufz.de (A.V.N.); 2Center for Microbial Electrochemical Systems (CEMES), Advanced Water Management Center, The University of Queensland, Brisbane QLD 4072, Australia

**Keywords:** *Pseudomonas putida*, Microbial electrochemical technology, bioelectrochemical system, anoxic phenotype, anaerobic cultivation

## Abstract

Industrial fermentation in aerobic processes is plagued by high costs due to gas transfer limitations and substrate oxidation to CO_2_. It has been a longstanding challenge to engineer an obligate aerobe organism, such as *Pseudomonas putida*, into an anaerobe to facilitate its industrial application. However, the progress in this field is limited, due to the poor understanding of the constraints restricting its anoxic phenotype. In this paper, we provide a methodological description of a novel cultivation technology for *P. putida* under anaerobic conditions, using the so-called microbial electrochemical technology within a bioelectrochemical system. By using an electrode as the terminal electron acceptor (mediated via redox chemicals), glucose catabolism could be activated without oxygen present. This (i) provides an anoxic-producing platform for sugar acid production at high yield and (ii) more importantly, enables systematic and quantitative characterizations of the phenotype of *P. putida* in the absence of molecular oxygen. This unique electrode-based cultivation approach offers a tool to understand and in turn engineer the anoxic phenotype of *P. putida* and possibly also other obligate aerobes.

## 1. Introduction

*Pseudomonas putida* (*P. putida*) is a promising industrial host for the production of harsh chemicals due to its resilience against environmental stresses (e.g., solvents) that will quickly diminish other industrial microorganisms [1,2,3,4]. This family, however, comprises obligate aerobes with oxygen exclusively used as the terminal electron acceptor in nature. Compared to anaerobic processes, oxygen-dependent metabolism can obtain higher energy yields, a high metabolic turnover and product purity [5,6], but it also suffers significantly from the comparatively low product yield and high operating cost for heat removal and oxygen supply [7,8]. These limitations pose a strong economic barrier for the industrial application of *P. putida*, as well as other obligate aerobic strains and limits feasible reactor sizes.

To overcome this bottleneck, it has been attempted to introduce a non-oxygen-dependent route into *P. putida* for electron balance and thus to create an anaerobic mutant of *P. putida*. Nikel et al. [9] engineered a fermentative pathway towards acetate and ethanol into *P. putida* KT2440 and another group reported an approach of introducing a nitrate/nitrite-based respiration pathway in the same strain [10]. Both systems were active in the recombinant strains; however, the mutants only showed limited metabolic activity and were not able to grow without oxygen. Identifying further engineering targets to improve those strains is urgently required but is extremely difficult due to the poor knowledge of the anaerobic constraints of an obligate aerobe (not limited to *P. putida* only) to date. It is difficult to identify the key metabolic pathways (i.e., the potential targets for strain engineering) since hardly any metabolic activity of *P. putida* is retained once oxygen is removed from the medium. A novel cultivating method that could (even partially) activate anoxic metabolism of *P. putida* would allow quantitative systems biology approaches to identify potential targets.

Over the last two decades, a technique termed microbial electrochemical technology has gained increased interest. The fundamental principle is to use an electrode as the electron acceptor/donor for microbial redox metabolism. Such a reactor is commonly named a bioelectrochemical system (BES) [11]. The ability to use a solid phase as electron acceptor via direct or indirect electron transfer route has been found for many anaerobic or facultative anaerobic microorganisms such as *Shewanella* [12], *Geobacter* [13], *Pseudomonas aeruginosa* [14] and more recently even gram-positive *Listeria* [15].

In this paper, we present a detailed protocol for the cultivation of *P. putida* in a BES reactor without oxygen as well as the general results to be expected. The redox power from an anode and a redox mediator can successfully drive the anoxic sugar metabolism of *P. putida* which then allows further quantitative physiology characterizations. This electrode-based cultivation approach provides a unique platform to quantitatively assess the anaerobic phenotype of *P. putida* cells, which cannot be achieved by any other technique so far. We believe that this lays the basis for rationally developing an anaerobically growing *P. putida*, and moreover, similar approaches may also be transferred to cultivate other obligate aerobic microbes without oxygen present.

## 2. Experimental Design

The basic principle of this cultivation protocol is applying an anode as the final electron acceptor for cellular redox metabolism. However, *P. putida* genetically have neither the extracellular iron respiration system like *Shewanella* and *Geobacter* for direct electron transfer nor the pyocyanin biosynthesis pathway existing in *Pseudomonas aeruginosa* for indirect electron transfer. Thus, to establish an electron flow between the cells and anode, an artificial redox mediator has to be added in the medium as an electron shuttle. Thermodynamically, the chemical should have a positive enough redox potential to withdraw the electrons from cellular electron transport system, and the threshold was found to be 0.207 V vs standard hydrogen electrode (SHE) [16].

Anaerobic growth of *P. putida* is not feasible so far, which is in fact the ultimate target of developing this protocol. Therefore, an aerobic pre-culture preparation process is required to obtain enough biomass for the BES cultivation and the subsequent quantitative physiology characterization. Figure 1 presents the general workflow of this protocol.

### 2.1. Reagents

Potassium ferricyanide (Sigma-Aldrich, Munich, Germany; Cat. no. 244023)LB-agar plate: tryptone 10 g/L, yeast extract 5 g/L, NaCl 10 g/L and agar 17.5 g/LTrace solution: FeCl_3_·6H_2_O 1.5 g/L, KI 0.18 g/L, H_3_BO_3_ 0.15 g/L, CoCl_2_·6H_2_O 0.15 g/L, MnCl_2_·4H_2_O 0.12 g/L, Na_2_MoO_4_·2H_2_O 0.12 g/L, ZnSO_4_·7H_2_O 0.12 g/L, CuSO_4_·5H_2_O 0.03 g/L, EDTA (acid form) 10 g/L, and NiCl_2_·6H_2_O 0.023 g/L.Potassium chloride solution: saturated concentration.DM9 medium: Na_2_HPO_4_ 6 g/L, KH_2_PO_4_ 3 g/L, NH_4_Cl 1 g/L, MgSO_4_·7H_2_O 0.10 g/L, 1 mL/L of CaCl_2_·2H_2_O stock solution (15 g/L), 1 mL/L of trace solution and glucose 5 g/L.Anode buffer (AB): Na_2_HPO_4_ 6 g/L, KH_2_PO_4_ 3 g/L, NH_4_Cl 1 g/L, MgSO_4_·7H_2_O 0.10 g/L, 1 mL/L of CaCl_2_·2H_2_O stock solution (15 g/L), 1 mL/L of trace solution and glucose 1.5 g/L.Cathode buffer (CB): Na_2_HPO_4_ 6 g/L, KH_2_PO_4_ 3 g/L and NH_4_Cl 1 g/L.

### 2.2. Equipment

Potentiostat (Bio-logic science instrument, Seyssinet-Pariset, France; Cat. no.: VSP)Heating/refrigerated circulators (Julabo, Seelbach, Germany, Cat. no.: 9352506, 9312640)Magnetic stirring plate (LLG labware, Meckenheim, Germany; Cat. no.: 6.263.440)Oxygen sensor (PreSens, Regensburg, Germany; Cat. no.: OXY-4 mini)Gas flow meter (Cole-Parmer, Wertheim-Mondfel, Germany; Cat. no.: GZ-03216-08)Incubator (Infors AG, Basel, Switchzerland; Cat. No.: Multitron)Centrifuge (Eppendorf, Hamburg, Germany; Cat. no.: 5810R)Bioelectrochemical cells (custom-made, see Section 2.3 for details)

### 2.3. Bioelectrochemical System Configurations

The locally-made reactor is designed based on the following considerations: (i) it is sealable and autoclavable to prevent contamination; (ii) it provides stable and defined conditions for the working chamber (anodic chamber) where the cells will be cultivated; (iii) it has sufficient working volume for multisampling and further analytics; and (iv) it is compatible for external plug-in sensors, e.g., pH, pO_2_, redox, etc.

In brief, we designed a double-jacketed cylinder-like glass reactor covered with polyether ether ketone (PEEK)-made lids and plugs. The working chamber is separated from the counter chamber (cathodic chamber) by an ion exchange membrane. The schematic diagram of the reactor is shown in Figure 1 and Figure 2 and described as below (the same order as numbered in Figure 1 step 2):**Glass vessel:** working chamber, double-jacketed cylinder-like vessel with DN60 flat flange as the open-top (DWK life sciences GmbH, Wertheim/main, Germany; Cat. no.: 21035340).NOTE: Two open ports of GL14 screw size connected to the out-jacket chamber are for temperature control, and the third open port of the same size connected to the inner working chamber is for inoculation.**Lid:** 100 mm diameter, fit DN60 flat flange and the corresponding quick release clamp (DWK life sciences GmbH, Wertheim/main, Germany; Cat. no.: 10464691). Seven open ports were designed on the lid for plugins, while six of them have the inner diameter of 12 mm and the left (in the center) is of 2-mm diameter for working electrode cable.NOTE: All the measuring and controlling units are plugged into the working chamber through the lid with screw plugs sealed with O-rings. The lid is made from polyether ether ketone (PEEK) material because of its stability and resistance against thermal, chemical and climate stresses.**Screw plugs**: made from PEEK for the seven open ports on the lid. Four different designs are used depending on the purpose: (i) plug with big hole (12 mm) in the center for glass tubes and sensors; (ii) plug with no hole used as sealing stoppers for sparing ports; (iii) plug with small hole (2 mm) only for working electrode cable; iv) plug with four small holes (3.2 mm each) for gassing, sampling and feeding.**Bridging tube**: a glass tube (out diameter of 12 mm) with the bottom-end sealed with porous glass frit (pore size of < 1µm).NOTE: Not essential, but quite beneficial for two reasons: (i) minimize the voltage drop; (ii) prevent reference electrode (Als, Tokyo, Japan; Cat. no.: 013503) from being contaminated. The glass frit could also be fixed via heat shrink PTFE tube as well, but this always falls off after autoclave.**Counter electrode compartment:** a hollow glass tube (out diameter of 12 mm) with a GL14 screw open end.NOTE: it is separated from the working chamber via cation exchange membrane using GL14 open-top cap and O-ring.**Condensers:** glass condenser with the flat neck of 12 mm diameter, the top open-end facing outside the reactor is sealed with an autoclavable membrane filter (0.22 µm pore size).NOTE: it is essential for balancing the reactor pressure and reducing water loss due to gas sparging.**Anode:** pretreated carbon cloth of 5 × 5 cm dimensions.
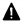
**NOTE: KEY** component of the whole system. The raw carbon cloth is quite hydrophobic on the surface and thus needs to be pretreated prior to use. The pretreatment method should increase hydrophilicity of the carbon cloth and also give a high overpotential for oxygen evolution. The successful methods tested are either soaking overnight (about 16 h, 40 °C, 200 rpm of shaking) or running cyclic voltammetry for 100 cycles in 2 mM cetrimonium bromide (CTAB) solution (Appendix A). Identical performance can be observed for both approaches. The working potential window, defined as no significant oxygen evolution observed (otherwise oxygen would be produced through electrochemically oxidation of H_2_O), for the treated carbon cloth can be over 0.8 V (vs SHE), which is sufficient for oxidizing the mediators used (discussion below). The unsuccessful pretreatment method tested was electrolysis in acids, which gave a strong background noise from the blank electrode (see Appendix A).**Membrane:** cation exchange membrane (Membranes international INC., Ringwood, NJ, USA; Cat. no.: CMI-7000), 12-mm diameter disc.NOTE: It is used to seal the bottom end of the counter electrode compartment (the one plugged into the working chamber) to separate it from the working chamber, using a GL14 open-top cap and o-ring fixed on the counter electrode tube. Other types of membranes can also be used but the pH gradient cross the membrane and gas permeability need to be considered.**Electric connecting wire**: titanium wire of 0.5-mm diameter (Advent, Oxford, England; Cat. no.: TI5555).NOTE: Titanium is selected for electrode connections, because it is nontoxic to microorganisms and highly stable against the electric, chemical and biologic stress.**Cathode**: stainless steel mesh (Membranes international INC., Ringwood, NJ, USA; Cat. no.: CMI-7000).NOTE: Other electric-conductive materials can also be used, such as platinum, carbon or graphite. In general, an electrode with low overpotential for hydrogen evolution or oxygen reduction will be recommended to improve the energy efficiency of the whole reactor. Changing cathode material will not affect the microbe cultivation in the working chamber. Some other reactors modified from a conventional stir-tank bioreactor may also be used if available [17,18]. Those reactors are more expensive but can give more precise control of such as the mass transfer in the liquid phase, which can be used for further process engineering and optimization.

## 3. Procedure

Following the general workflow given in Figure 1, the detailed procedure of cultivating wild-type *Pseudomonas putida* is presented below. All media are sterilized by either autoclaving or membrane filtration, and all work should be done in a sterilized environment, unless otherwise stated.

### 3.1. Pre-Culture Preparation (Time for Completion: 3 Days, from Day 1 to Day 3)

NOTE: Maintain consistent procedures, especially considering the incubation times are essential to minimize batch to batch variations. Differences in the metabolic status of the precultures will increase variability in the BES performance.

Steak a *Pseudomonas putida* cryo stock on LB-agar plate.NOTE: Single colonies should be visible on the plate after incubation (i) to reduce the heterogeneity of colonies picked for liquid culture and (ii) to check and avoid the contamination on the plate.Incubate at 30 °C for 24–36 h.NOTE: longer incubation time will cause a longer lag phase for liquid culture growth.Pick up a single colony and inoculate in a baffled Erlenmeyer shaking flask containing DM9 medium.**CRITICAL STEP**: To improve sufficient oxygen supply, cotton stoppers or membrane vent caps (not silicon-based stoppers) should be used for the gas exchange (see Appendix A); moreover, the flasks should be filled with medium of <= 20% (identical amount for all batches) of the nominal volume of the flask.Incubate the liquid culture overnight (15–16 h);Collect the cell pellets by centrifugation at 30 °C, 7000 g, 10 min;Resuspend the cell pellets in 15 mL AM buffer and suck the solution in a syringe for immediate injection (Section 3.3)NOTE: Precultures should be timed in a way that cells can be harvested once the reactors are ready for injection (see below)

### 3.2. Reactor Assembly (Time for Completion: 2 Days, from Day 1 to Day 2)

NOTE: This step is running simultaneously with step 3.1. Normally, it is to assemble the reactor after step 1 and then do the autoclave overnight (simultaneously with step 2).

7.Assemble the BES reactor accordingly, as the final picture shows in Figure 2.NOTE: The sampling tube on the four-port plug should reach close to the stir bar, while the gassing tube should be above the liquid phase (it is only to gas the headspace). A new working electrode should be used for each batch.8.**OPTIONAL STEP**: plug pH, pO_2_ or redox sensors into the two spare ports on the lid if necessary.NOTE: Some autoclavable pH sensors were found to be quite sensitive to the electric field after autoclaving. The value can be shifted dramatically (up to 1 pH unit), or become unresponsive after applying the desired potential on the working electrode. This issue can be avoided by sterilizing the pH sensor with 80% ethanol instead of autoclaving.9.Fill 50 mL CB buffer into the working chamber.NOTE: For autoclaving purpose. Not AB buffer, because glucose is not autoclavable together with salt.10.Fill the counter electrode tube with CB buffer until full.11.Fill the bridge tube with saturated KCl solution until full.12.Check the gas tightness of the assembled reactor.
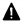
**CRITICAL STEP**: The reactor should be gas-tight to prevent the entry of oxygen from the outside atmosphere into the reactor and to minimize the risk of contamination during operation. Procedure to check the gas tightness: connect the gassing tube to gas line, block the condenser and open the sampling tube. The filled-in buffer should be pushed out of the sampling tube if the reactor is air tight. Otherwise, please check the O-rings for each connection.13.Autoclave the assembled reactor.NOTE: reference electrode and the stainless-steel counter electrode should not be autoclaved. If necessary, carbon cloth and stainless-steel electrode can be cleaned by two-step sonication in solvent (e.g., ethanol or methanol) and sterile distilled water respectively. Reference electrode cannot be sonicated and should only be washed gently with ethanol and water, or be maintained following the manufacturer’s instructions.

### 3.3. BES Cultivation (Time for Completion: x Days, from Day 2 until Experiment Terminated)

14.Transfer the autoclaved reactor to a sterilized clean bench.15.Remove the liquids inside the working chamber and refill 300 mL AB buffer;
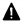
**CRITICAL STEP**: It is important to check that there is no bubble blocking the cation exchange membrane (Nr. 6 in the Figure 2 B) while filling the medium. This is the first of the two most common reasons that will break the electric circuit.16.Fix the reference electrode on the bridging tube.
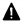
**CRITICAL STEP**: Check the buffer level inside the reference electrode and refill with saturated KCl solution if necessary. This is the second most common reason that will break the circuit. In addition, some KCl crystals should be visible inside the reference electrode and bridging tube, to guarantee the saturation during the experiment.17.Place the counter electrode inside the counter electrode compartment.18.Open the gassing tube and connect to a sterile syringe membrane filter (0.22 µm pore size, PTFE).19.Transfer the reactor to the working bench and place it on the magnetic stir plate, setup the stirring speed of 400 rpm.NOTE: Stirring speed can be adjusted but has to be consistent for the all batches, unless mass transfer parameters are to be studied. High stirring speed results in a high mass transfer rate in the system.20.Connect the double-jacketed chamber to a recirculating heating water bath (set to 30 °C).NOTE: It is important to check the temperature drop while running multiple reactors in parallel. A maximum number (normally 4–6) of reactors should be determined for a given heating circulator system.21.Connect the condenser to a recirculating chiller (set to 6 °C).22.Connect the gassing port to the nitrogen gasline (flow rate set up to be 20–30 mL/min).
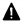
**CRITICAL STEP**: Values chosen in step 21–22 are critical to minimize evaporation of water during operation. For the conditions given above, we measured ~0.09mL water would be lost per hour. This value is essential for the quantitative evaluation (e.g., carbon and electron balance) of the whole system.23.Connect the working electrode, counter electrode and reference electrode to the potentiostat.24.Start the potentiostat and apply a chronoamperometry method to the BES reactor with the following parameters: working electrode potential (E_we_) of “0.5 V” vs Ag/AgCl/KCl_sat_, current (I) recording of every dI of 100 µA and every dt of 10 min.NOTE: working electrode potential needs to be high enough to efficiently oxidize the mediator, which can be determined by cyclic voltammetry method; 0.5 V is determined for ferricyanide. This value needs to be adjusted when changing mediators, for instance 0.3 V for [Co(bpy)_3_]Cl_3_).25.Record current for 5–8 h for the fresh AB medium (over the day time of day 2), and a flat background signal should be achieved before the next step;NOTE: Quality control I: The current recorded on the potentiostat for blank AB medium should be around 0 µA fluctuated within a maximum of ±10 µA range. Larger noise background is likely coming from the improper electric connections and/or contaminants present in the buffer or working electrode surface, which needs to be checked.26.Inject a final concentration of 1 mM ferricyanide (about 3–5 mL of stock solution) into the working chamber, recording the current for 14–16 h (overnight from day 2 to day 3), and a flat background signal should be achieved before the next step.NOTE: Quality control II: The signal should most likely fluctuate between 0–10 µA, with a maximum up to 20 µA. Otherwise, the system needs to be checked.27.Inoculate the BES reactor with the biomass from 3.1, record current until manually stopped;28.**OPTIONAL STEP**: Check the dissolved oxygen level with plugged-in pO_2_ sensor, and the value should be 0.29.**OPTIONAL STEP**: Control/monitor the pH value of the system.NOTE: While multi-BES reactors are controlled in parallel, one possible issue for the pH system is the cross-interference between different reactors due to the electrical connection of the systems through the potentiostat. The change of applied electrode potential in one reactor can possibly affect the pH sensor in another reactor. Solutions to this issue can be galvanically isolating different pH channels or choosing optical pH measurements.30.Sample the reactor once a day (or more frequently) along the batch and collect supernatant and/or cell pellets depending on the analytic purposes;31.Apply quantitative approaches for analyzing genotype, phenotype and/or metabolism.

## 4. Expected Results

The described protocol has been experimentally validated for cultivating different *P. putida* strains including type-strains as well as recombinant strains. Typical raw results for a short-term BES batch without pH control using the type-strain *Pseudomonas putida* F1 are presented in Figure 3.

The increasing current detected by potentiostat for biotic condition confirms the establishment of electron flux from the cells to electrode. Glucose, as being the only useable carbon and electron source present in the system, is consumed by the cells, further demonstrated by the decrease of glucose concentration in the BES reactor quantified by HPLC. Accompanying the glucose consumption, organic acids are accumulated in the broth and thus, pH is dropping. The major product is 2-ketogluconic acid with the carbon molar yield of > 90%, and acetic acid is the only significant by-product. No significant glucose oxidation could be detected if there is no mediator and/or electric power applied.

As expected, the biomass density is also decreasing, and no anaerobic growth can be observed. The sugar consumption rate is much lower (only < 5%) compared to the value reported for aerobic cultivations. Moreover, the products indicate the carbon metabolic activity is mainly constrained to the periplasmic space, while the cytoplasmic central metabolism is only partially activated (to produce acetic acid). Nevertheless, a complete anaerobic consumption of sugar is achieved for an obligate aerobe, providing a system to quantitatively investigate and identify the metabolic constraints limiting the anaerobic metabolism of *P. putida* using systems biology approaches

## Figures and Tables

**Figure 1 mps-02-00026-f001:**
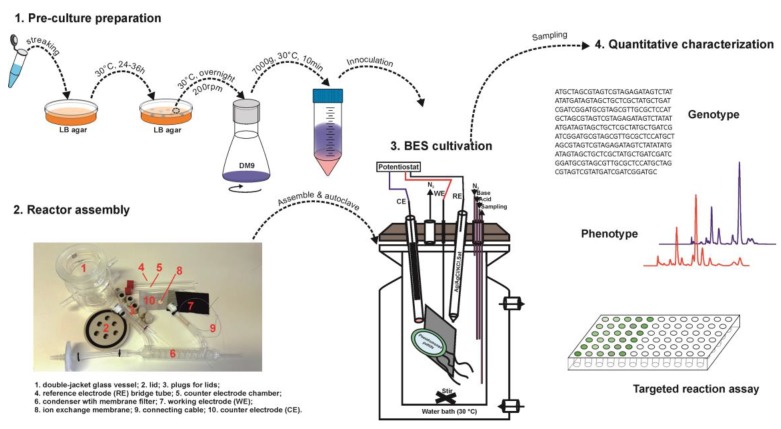
The general working flow for the cultivation and characterization of *Pseudomonas putida* in bioelectrochemical system.

**Figure 2 mps-02-00026-f002:**
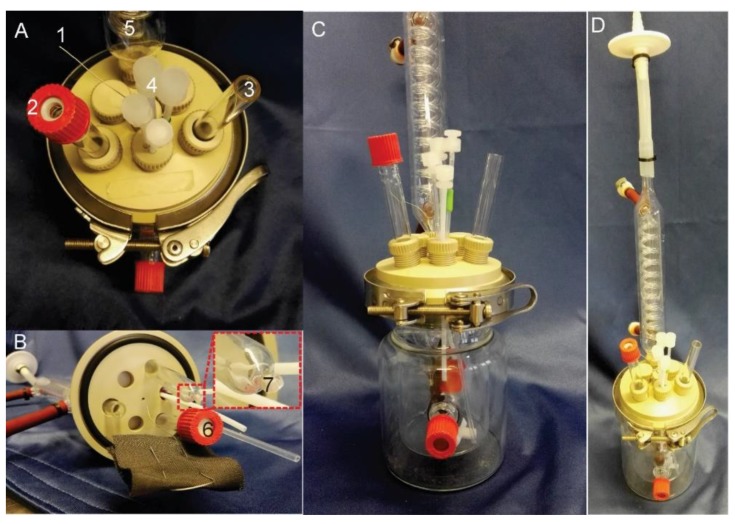
Pictures of the assembled BES reactor. (**A**) top-view; (**B**) bottom view of the lid; (**C**) and (**D**) overview from the side. 1, working electrode electric wire; 2, top-end of the bridging tube; 3, top-end of the counter electrode compartment; 4, four-port plug for gassing, sampling and feeding; 5, condenser; 6, bottom-end of the counter electrode compartment, sealed by an ion exchange membrane; 7, bottom-end of the bridging tube, sealed by porous glass frit (pore size of < 1 µm).

**Figure 3 mps-02-00026-f003:**
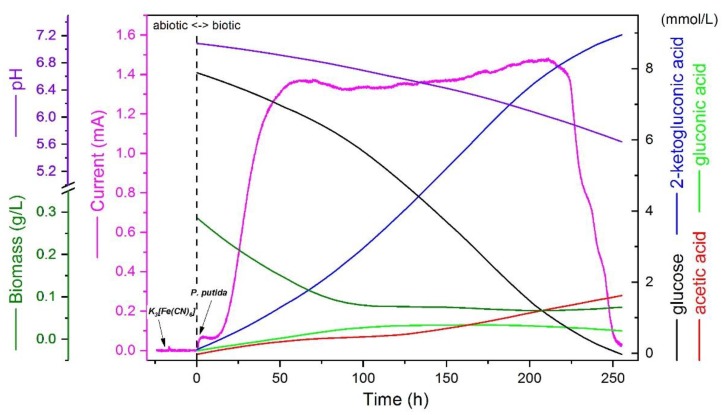
Typical raw results of the short-batch BES cultivation of *Pseudomonas putida* F1 with glucose as the sole carbon source (averaged from triple reactors). Ferricyanide of 1mM is used as the mediator and the working electrode potential is set at 0.5 V vs Ag/AgCl/KCl_sat_. Right *Y* axis indicates the concentrations of different compounds in the broth quantified by HPLC. They need to be adjusted with a concentrating factor because of the water loss caused by gas flushing (0.09 mL/h) while conducting carbon and electron balance (both balanced for the presented data) based on those data.

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
