# Peer review of "Characterizing the Anoxic Phenotype of Pseudomonas putida Using a Bioelectrochemical System"

_mps, 2019, doi:10.3390/mps2020026_

Round 1

Reviewer 1 Report

Characterizing the anoxic phenotype of Pseudomonas putida using a bioelectrochemical system

This manuscript describes a method to switch pseudomonas putida from aerobic metabolism to anaerobic metabolism using redox mediators and electrodes. This opens the possibility for controlled production of chemicals under anaerobic conditions and perhaps also to steer the production through changing the metabolic pathways.

The protocol is well described and I’m very curious to find what implications and results this protocol will generate.

The only remark I have is that the anion exchange membrane mentioned in the materials is a cation exchange membrane: CMI-7000 The same reference number is used for the cathode, which I believe is a mistake?

Author Response

Yes, it was a typo error at line 150. We corrected the "anion exchange membrane" to be "cation exchange membrane". See line 150 in the revised manuscript.

Reviewer 2 Report

Dear Authors,

in my opinion your work is very interesting and well organized. Please consider the following points for redrafting purposes.

-) please define the following acronyms before using them: SHE (r.70), IR (r.128);

-) r.218: I think you could add some hint about the alternative electrodes cleaning process before their application to the reactor;

-) r.303: please re-write the following sentence to clarify its meaning: "and thus enable the feasibility to further apply systems biology approaches to"

Author Response

Changed. line 70 "... standard hydrogen electrode (SHE)", and we change the "IR drop" in line 128 to "voltage drop" to make it clearer.

Added. line 218-222 in the revised manuscript: "... While necessary, carbon cloth and stainless-steel electrode can be cleaned by two-step sonication in solvent (e.g. ethanol or methanol) and distilled water respectively. Reference electrode cannot be sonicated and should only be washed gently with ethanol and water, or be maintained following the manufacturer’s instruction. "

Changed. the sentence is rephrased to be "... thus it would be feasible to further quantitatively investigate ... using systems biology approaches".

Round 2

Reviewer 1 Report

ok

Author Response

Thanks for the endorsement.